# A nanobody suite for yeast scaffold nucleoporins provides details of the nuclear pore complex structure

Sarah A. Nordeen [1], Kasper R. Andersen [1], Kevin E. Knockenhauer[1], Jessica R. Ingram[2], Hidde L. Ploegh [2] & Thomas U. Schwartz [1]✉

Nuclear pore complexes (NPCs) are the main conduits for molecular exchange across the nuclear envelope. The NPC is a modular assembly of ~500 individual proteins, called nucleoporins or nups. Most scaffolding nups are organized in two multimeric subcomplexes, the Nup84 or Y complex and the Nic96 or inner ring complex. Working in *S. cerevisiae*, and to study the assembly of these two essential subcomplexes, we here develop a set of twelve nanobodies that recognize seven constituent nucleoporins of the Y and Nic96 complexes. These nanobodies all bind specifically and with high affinity. We present structures of several nup-nanobody complexes, revealing their binding sites. Additionally, constitutive expression of the nanobody suite in *S. cerevisiae* detect accessible and obstructed surfaces of the Y complex and Nic96 within the NPC. Overall, this suite of nanobodies provides a unique and versatile toolkit for the study of the NPC.

[1] Department of Biology, Massachusetts Institute of Technology, Cambridge, MA, USA. [2] Boston Children's Hospital and Harvard Medical School, Boston, MA, USA. ✉email: tus@mit.edu

The hallmark of the eukaryotic cell is a complex endo-membrane system of organelles that compartmentalize specific functions within the cell. One of the largest of these organelles is the nucleus, which stores the genetic material and is the site of replication, transcription, and ribosome synthesis. Soluble transport of molecules into and out of the nucleus occurs solely through nuclear pore complexes (NPCs), 40–120 MDa ring-like channels that perforate the inner and outer membranes of the nuclear envelope (NE) (Fig. 1a). Roughly 30 nucleoporins (nups) contribute to the modular, eightfold symmetric assembly of subcomplexes that comprise the NPC[1–4]. One of the main structural subcomplexes is the Nup84 or Y complex. In *S. cerevisiae*, the 575 kDa Y complex has seven components, many of which are essential or produce severe phenotypes when deleted[5–7]. The Y complex structure consists of two short arms that meet a long stalk at a central triskelion-like hub[8–10] (Fig. 1b). The other main structural component of the NPC is the hetero-meric Nic96 or inner ring complex[11]. This ~0.5 MDa complex occupies the inner ring of the NPC and anchors the trimeric Nsp1 complex, which contains three phenylalanine-glycine (FG)-nups that are essential for maintaining the permeability barrier[11–16]. Both Y complex nups (Nup84, Nup85, and Nup145C) and Nic96 have an ancestral coatomer element 1 (ACE1) fold, conserved across the NPC and COPII vesicle coats[17,18]. The tripartite fold consists of crown, trunk, and tail modules and the interfaces between each module act as somewhat flexible joints. How this flexibility affects the NPC assembly is still unclear.

Its size, flexibility, and membrane interactions pose challenges for the elucidation of the structure of the NPC. But only with detailed structural information will we obtain mechanistic insight into the many functions of NPCs. To arrive at a structure, different labs approach the problem by either a bottom-up or top-down approach. For the bottom-up approach, the many modular structural assemblies that make up the NPC have been broken down into further sub-assemblies which are then studied primarily by X-ray crystallography. Over the past decade, many of the structural elements have been characterized (reviewed in refs. [2,19]). We now have a complete composite model for the Y complex from *S. cerevisiae*[20]. For the top-down approach, cryo-electron tomography (cryo-ET) has been used to visualize whole NPCs while still embedded in the nuclear membrane or after detergent extraction. Recent studies have yielded maps of the entire NPC at 2–5 nm resolution for human, *Xenopus laevis*, and *S. cerevisiae*[21–24]. The resolution gap between the top-down and bottom-up approaches has narrowed. Along with multiple stoichiometry studies, docking of the many crystal structures of nups into the cryo-ET maps has been attempted[22,23,25–28]. For *S. cerevisiae*, the cryo-ET map allows placement of Y complexes into the density, resulting in a model that contains a total of 16 copies per NPC[24]. This model consists of two eight-membered rings, one each on both the cytoplasmic and nucleoplasmic faces of the NPC[23,24]. The Y complexes are arranged in a head-to-tail manner, with the main interface mediated by Nup120 and Nup133[29]. On the cytoplasmic face, the Nup82 complex anchors to the Y complex via Nup85[30]. Thirty-two copies of Nic96 are nestled tightly into the inner ring complex along with Nup192, another scaffolding nup[23]. While these studies improved our understanding of the overall architecture of the NPC, the resolution of these cryo-ET maps does not reveal secondary structure. This leaves room for further interrogation and improvement. We aimed to create a set of tools that will aid in studying the complicated NPC assembly in more detail.

Here we describe a nanobody library comprising 12 unique nanobodies to the Y complex and Nic96 from *S. cerevisiae*. Nanobodies are single-domain (VHH) antibody fragments derived from camelid heavy-chain only antibodies. Nanobodies are excellent tools both in vitro and in vivo, as they are small (~14 kDa), easily purified from *E. coli*, easily modified with fluorophores, and typically have nanomolar binding affinities[31,32]. The library consists of nanobodies that bind to each of the 6 conserved nups in the Y complex[8]. We describe their in vitro binding characteristics using bio-layer interferometry, size-exclusion chromatography (SEC), and, in several cases, their nup-bound structures by X-ray crystallography. We show the effects of nanobody expression in vivo and how these results suggest accessible and inaccessible surfaces within the assembled NPC. Together, this work provides a toolkit for studying the scaffold of the NPC and uncovers details of the NPC structure in vivo.

## Results

**A nanobody library to the Y complex and Nic96.** In order to generate nanobodies specific to NPC scaffold nups, we separately immunized alpacas with recombinantly purified full-length Y complex and Nic96. We then selected nanobodies by phage display using single nups or subassemblies of the Y complex as targets (Fig. 1b). This allowed us to obtain nanobodies that cover both short arms, the hub, and the long stalk of the Y complex. We thus compiled a set of twelve nanobodies that bind Y complex nups and Nic96 (Fig. 1b, c). The library covers a wide range of sequence space (Fig. 1c). Both, the sequences and lengths of complementarity determining regions (CDRs) 1 and 2, are more similar than CDR3 across the library. However, there is no common CDR across the set. The greatest differences arise in CDR3, which varies from 9 to 24 residues. This wide deviation in length is attributed to the vast genetic diversity contributed by the immunized alpaca, rather than the typically shorter and invariant lengths used in in vitro selection methods[33,34].

After selection, we obtained two nanobodies (VHH-SAN6 and 7) that recognize a truncated Y complex hub construct (Nup120 and Nup85 C-terminal domains, full-length Nup145C-Sec13) (Fig. 1b). In order to identify the nups these nanobodies target, we first tested binding by SEC with full-length nups (Nup120, Nup85, and a fusion construct of Nup145C-Sec13[35]). We found that both bound Nup145C-Sec13. To further narrow down the binding site, we conducted a second SEC experiment using only Sec13 fused to the Nup145C insertion blade[35] and found that VHH-SAN7 recognizes Sec13 rather than Nup145C (Supplementary Fig. 1). VHH-SAN6 did not bind Sec13 alone, meaning that it recognizes Nup145C. Sec13 is tightly packed within the hub of the Y complex, so we tested the ability of VHH-SAN7 to bind the assembled Y complex hub. When pre-incubated with Nup120-Nup85-Seh1-Nup145C-Sec13, VHH-SAN7 co-elutes as a heterohexameric complex (Supplementary Fig. 1). This experiment suggests that VHH-SAN7 binds Sec13 in the context of the assembled Y complex.

Three nanobodies (VHH-SAN1/2/3) in the library recognize Nup85. As an ACE1 nup, we examined whether these nanobodies bind to one of three modules (crown, trunk, tail) within the domain. By SEC analysis we found that all three nanobodies bound Nup85$_{trunk-crown}$-Seh1 (Supplementary Figs. 2, 3). We then tested the binding of each nanobody to Nup85$_{crown}$, which formed a stable complex with VHH-SAN2 and 3, but not VHH-SAN1 (Supplementary Fig. 2). Binding data suggest that VHH-SAN2 and 3 recognize distinct, non-overlapping epitopes, as we observe a heptameric complex of Nup85$_{trunk-crown}$-Seh1-VHH-SAN2/3 by SEC (Supplementary Fig. 2). We verified that VHH-SAN1 bound Nup85 by observing no complex formation with Seh1 alone (Supplementary Fig. 3).

**Nanobodies bind with varying kinetics, but strong affinities.** In order to characterize the binding kinetics of our nanobody

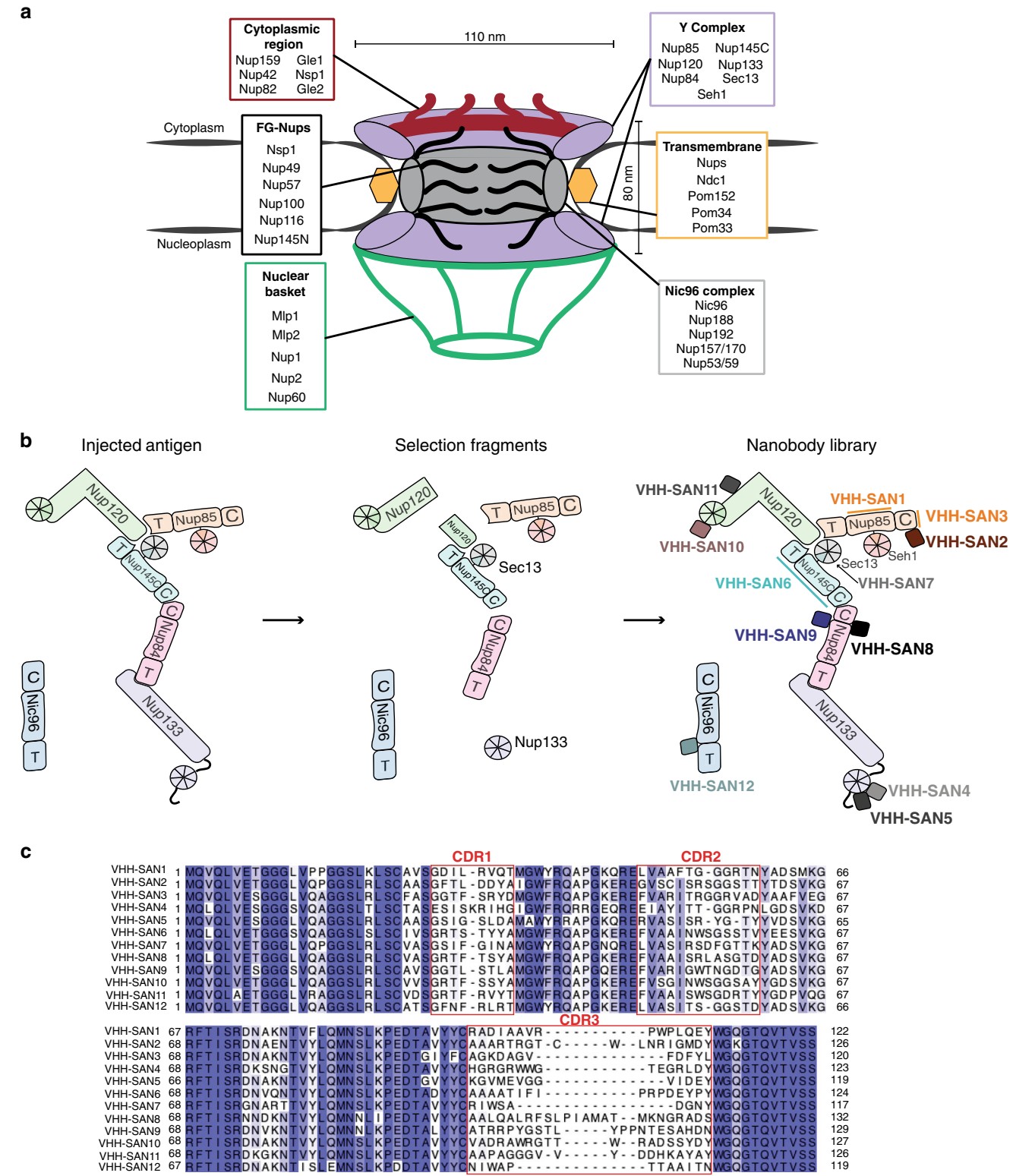

**Fig. 1 A nanobody library to scaffold nucleoporins. a** Schematic of the NPC assembly and classification of yeast nups. Each subcomplex or type of nup is listed together in a box and the relative position within the NPC assembly is indicated. **b** Schematic of the Y complex and Nic96 used for alpaca immunization, selection by phage display, and resulting nanobody library. Elements of ACE1 fold proteins are indicated: T—tail and C—crown adjacent to the central trunk element. **c** Sequence alignment of the nanobody library, colored by percentage identity. Variable complementarity determining regions (CDRs) are indicated.

library, we employed bio-layer interferometry (BLI). We affixed each nanobody with a C-terminal biotinylated Avi-tag[36] to streptavidin-coated biosensor tips. We assayed binding to each nanobody's respective nup target and observed a variety of binding kinetics across the nanobody library (Table 1, Fig. 2). The tightest binder, VHH-SAN3, dissociates very slowly resulting in a binding constant of ~14 pM. In fact, a slow dissociation rate applies to the majority of the library. A few nanobodies,

VHH-SAN2, 3, 8, and 9, have off rates less than $\sim 1 \times 10^{-4}\,\mathrm{s}^{-1}$ resulting in very tight equilibrium binding constants (14–170 pM). Of the set, VHH-SAN4 has the comparatively weakest affinity of ~230 nM, due to a much faster off rate than the other members the library and is the only nanobody that has an equilibrium binding constant of less than ~10 nM. Overall, the nanobodies bind very stably to their antigens.

**Mapping of nanobody epitopes by X-ray crystallography**. We mapped the epitopes of eight of the twelve nanobodies by solving the crystal structures of multiple nup-nanobody complexes. This enabled us to visualize the binding epitopes in greater detail. We solved the structures of nups that cover both short arms and the stalk of the Y complex: Nup85-Seh1-VHH-SAN2, Nup120$_{1-757}$-VHH-SAN10/11, and Nic96$_{186-839}$-VHH-SAN12. We used

VHH-SAN4/5 and VHH-SAN8/9 as crystallization chaperones for the previously uncharacterized Nup133$_{NTD}$ and Nup84-Nup133$_{CTD}$ complex structures, respectively (see the accompanying paper, https://doi.org/10.1101/2020.06.19.161133). VHH-SAN4 and five bind adjacent epitopes on the same face of the Nup133$_{NTD}$ β-propeller. Both, VHH-SAN4 and VHH-SAN5, contributed significant packing interfaces in the crystal lattice, facilitating high-resolution diffraction of a previously elusive target. In addition, both nanobodies rely primarily on CDR3 to recognize Nup133, with VHH-SAN4 making no contacts with either its CDR1 or 2. This is perhaps why VHH-SAN4 has the lowest binding affinity of the library, as CDRs 1 and 2 typically enhance binding strength[37]. Like Nup85, Nup145C, and Nic96, Nup84 has an ACE1 domain fold. Interestingly, both Nup84-specific nanobodies (VHH-SAN8 and 9) bind at the crown-trunk module interface, recognizing opposite faces of Nup84. It is not unusual for nanobodies to bind to a domain-domain interface[38]. While our attempts at co-crystallization of VHH-SAN1, 3, 6, and 7 with their nup targets failed, we were able to narrowly define their binding sites using SEC with smaller nup constructs.

We solved Nup85$_{trunk-crown}$-Seh1 in complex with VHH-SAN2 by molecular replacement (MR) using the structure of scNup85$_{1-564}$-Seh1 as a search model[18] (Fig. 3). From previous biochemical experiments, we expected that VHH-SAN2 binds the Nup85 crown module (Supplementary Fig. 2). After examining the Nup85 crown in the electron density map, we observed additional difference density, but not sufficient to encompass an entire nanobody (Supplementary Fig. 4). This made it difficult to place the nanobody by MR. We generated a VHH-SAN2 model using SWISS-MODEL[39] and manually placed it into the difference density map (Supplementary Fig. 4). We used a nanobody model that included the CDR loops, as these residues are most likely what make up the density closest to Nup85. After several rounds

| Table 1 Nanobody-binding affinities. | | | | |
|---|---|---|---|---|
| **Nanobody** | **Nucleoporin** | $K_D$ (M) | $k_{on}$ (M$^{-1}$s$^{-1}$) | $k_{off}$ (s$^{-1}$) |
| VHH-SAN1 | Nup85 | $5.8 \times 10^{-10}$ | $2.3 \times 10^{6}$ | $1.4 \times 10^{-3}$ |
| VHH-SAN2 | | $8.0 \times 10^{-11}$ | $1.0 \times 10^{6}$ | $8.1 \times 10^{-5}$ |
| VHH-SAN3 | | $1.4 \times 10^{-11}$ | $5.8 \times 10^{5}$ | $8.1 \times 10^{-6}$ |
| VHH-SAN4 | Nup133 | $2.3 \times 10^{-7}$ | $9.5 \times 10^{5}$ | $2.2 \times 10^{-1}$ |
| VHH-SAN5 | | $1.0 \times 10^{-8}$ | $2.8 \times 10^{5}$ | $2.8 \times 10^{-3}$ |
| VHH-SAN6 | Nup145C | $2.4 \times 10^{-10}$ | $5.3 \times 10^{5}$ | $1.3 \times 10^{-4}$ |
| VHH-SAN7 | Sec13 | $5.1 \times 10^{-10}$ | $1.3 \times 10^{6}$ | $6.5 \times 10^{-4}$ |
| VHH-SAN8 | Nup84 | $6.3 \times 10^{-11}$ | $3.5 \times 10^{5}$ | $2.2 \times 10^{-5}$ |
| VHH-SAN9 | | $1.7 \times 10^{-10}$ | $4.4 \times 10^{5}$ | $7.7 \times 10^{-5}$ |
| VHH-SAN10 | Nup120 | $3.8 \times 10^{-10}$ | $4.5 \times 10^{5}$ | $1.7 \times 10^{-4}$ |
| VHH-SAN11 | | $7.9 \times 10^{-10}$ | $9.8 \times 10^{5}$ | $7.8 \times 10^{-4}$ |
| VHH-SAN12 | Nic96 | $1.3 \times 10^{-9}$ | $1.8 \times 10^{6}$ | $2.4 \times 10^{-3}$ |

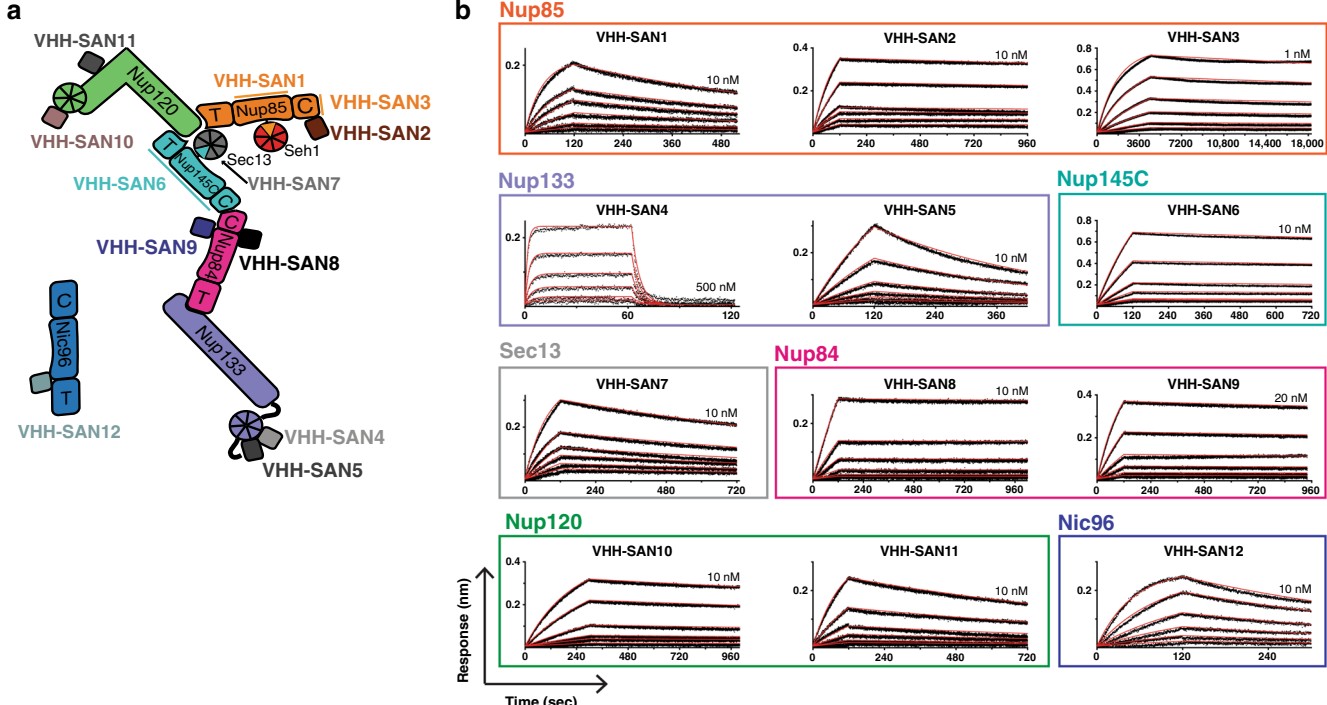

**Fig. 2 Bio-layer interferometry of nanobody-nup binding. a** Schematic of the Y complex and Nic96 with the nanobody library. **b** Binding curves showing association and dissociation kinetics for each nanobody-nup pair. Nanobodies with a biotinylated C-terminal Avi-tag were fixed (ligand) and nups were used as analytes. Curves were corrected for buffer background. Each set of curves is a twofold dilution series from the highest concentration listed on each plot. Experimental data are shown as black dotted lines with red lines indicating globally fitted curves. Axis labels for all plots are indicated in the bottom left.

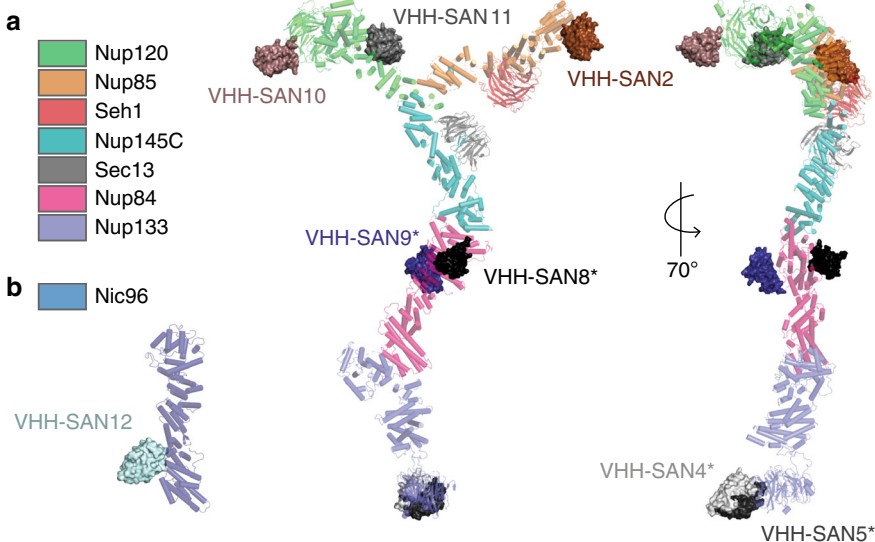

**Fig. 3 Structures of nanobodies complexed to the Y complex and Nic96. a** Structures of nanobody-nup complexes docked onto the composite structure of the Y complex. Nanobodies are shown as surfaces, the Y complex is shown as cartoon. VHH-SAN10/11 was solved in complex with Nup120$_{1-757}$. VHH-SAN2 was solved in complex with Nup85-Seh1$_{1-564}$. *Indicates structures described in ref. [20]. **b** Structure of Nic96 in complex with VHH-SAN12.

of positional refinement, the nanobody fit reasonably well into the density (Supplementary Fig. 4).

**VHH-SAN10 recognizes a flexible loop in Nup120.** We also solved the complex of Nup120$_{1-757}$-VHH-SAN10/11 by MR using the published scNup120$_{1-757}$ structure as a model[40]. After initial refinement, we observed additional density near residues 431–439 of Nup120, which we could attribute to a nanobody using MR. However, there was no obvious additional density for a second nanobody. Analysis of the crystals by SDS-PAGE showed that both nanobodies were present in the crystal (Supplementary Fig. 5). We then examined the crystal packing to see if there was space to accommodate the missing nanobody adjacent to unstructured loops. We hypothesized that if the nanobody was completely missing in the density, the epitope was also disordered in the crystal and therefore missing in the density as well. There are five unstructured loops in the β-propeller domain of Nup120 that neighbor solvent channels (Supplementary Fig. 5). A nanobody's dimension are roughly 35 Å × 15 Å × 15 Å. Only one large loop, residues 187–203, faces a solvent channel in the crystal that appeared sufficiently large to accommodate a nanobody. We hypothesized that the missing nanobody bound this loop and tested whether replacement of residues 187–203 with a flexible linker (GGSx5) would ablate binding by BLI. Indeed, we found that VHH-SAN10 no longer bound this Nup120 mutant, but as expected, VHH-SAN11 still recognized Nup120 (Fig. 4c). In a reciprocal experiment, we replaced residues 431–439 with a flexible linker (GSSx3) and tested VHH-SAN10 versus VHH-SAN11 binding. This experiment confirmed that VHH-SAN11, and not VHH-SAN10, binds to the 431–439 region (Fig. 4b).

Both VHH-SAN10 and 11 are interesting nanobodies, as they recognize relatively unstructured regions of Nup120. Typically, nanobodies recognize structured regions or clefts in their target, owing to their long CDR loops that prefer to insert into and along concave surfaces[38]. Regardless of this difference, both have very high binding affinities (Table 1). VHH-SAN10 and 11 add to a growing list of nanobodies that bind to short epitopes that function outside of a folded domain[41–44].

**VHH-SAN12 binds Nic96 between its trunk and tail modules.** In addition to the Y complex nanobodies, we identified a

nanobody, VHH-SAN12, that binds Nic96 of the inner ring complex. We solved the complex of Nic96$_{186-839}$ with VHH-SAN12 by MR with the published scNic96$_{186-839}$ structure as a template (Fig. 5a)[45]. Nic96$_{186-839}$ forms an elongated structure of 30 α-helices. Overall, the dimensions, shape, and ACE1 fold of Nic96 are identical to the previously described structures[45,46]. The N terminus is in the center of the protein that then zig-zags towards one end of the molecule. Helices α4–12 fold over themselves, forming the crown of the ACE1 domain, with α6–9 running perpendicular to the trunk helices of α13–21. The C-terminal helices α22–30 form the tail and zig-zag away from the trunk at an angle. VHH-SAN12 inserts its CDR loops 1 and 2 into the space between helices α20–21 and α22–25. Interestingly, VHH-SAN12 has one the shortest CDR3 of the library and CDR3 contributes little to the binding interface, which is unusual for a nanobody[37]. Even without this contribution, VHH-SAN12 has a high affinity for Nic96 (Fig. 2, Table 1).

Helices α20–25 delineate the trunk and tail interface of the ACE1 fold. In comparison to the previously solved structures, we observe a change in conformation. Aligning the tail domains of this structure and the structure from Jeudy et al.[45], we observe a kink that translates through the remainder of the molecule (Fig. 5b). VHH-SAN12 brings helices α20–21 and α22–25 slightly closer to each other, resulting in a ~19 Å shift between the two crown domains. This shift is slightly smaller (~10 Å) when comparing our structure to the one described in Schrader et al.[46]. While the overall conformation changes, the three modules of the ACE1 fold individually superpose very well. In comparing the crown domains, our new structure is much better defined (Fig. 5c). At the increased resolution (from 2.5 to 2.1 Å), we were able to complete Nic96 by including helix α9, and two loops including 30 additional residues. We speculate that the improvement of the data is due to the ability of the nanobody to stabilize Nic96 in one conformation. Either this is an effect of the nanobody alone, or it may be a combination of nanobody-binding paired with crystal packing.

**Several NPC nanobodies localize to the NE in vivo.** Having characterized this nanobody library in vitro, we asked whether these nanobodies would bind their targets in vivo and whether their expression would affect cellular fitness. We first put the

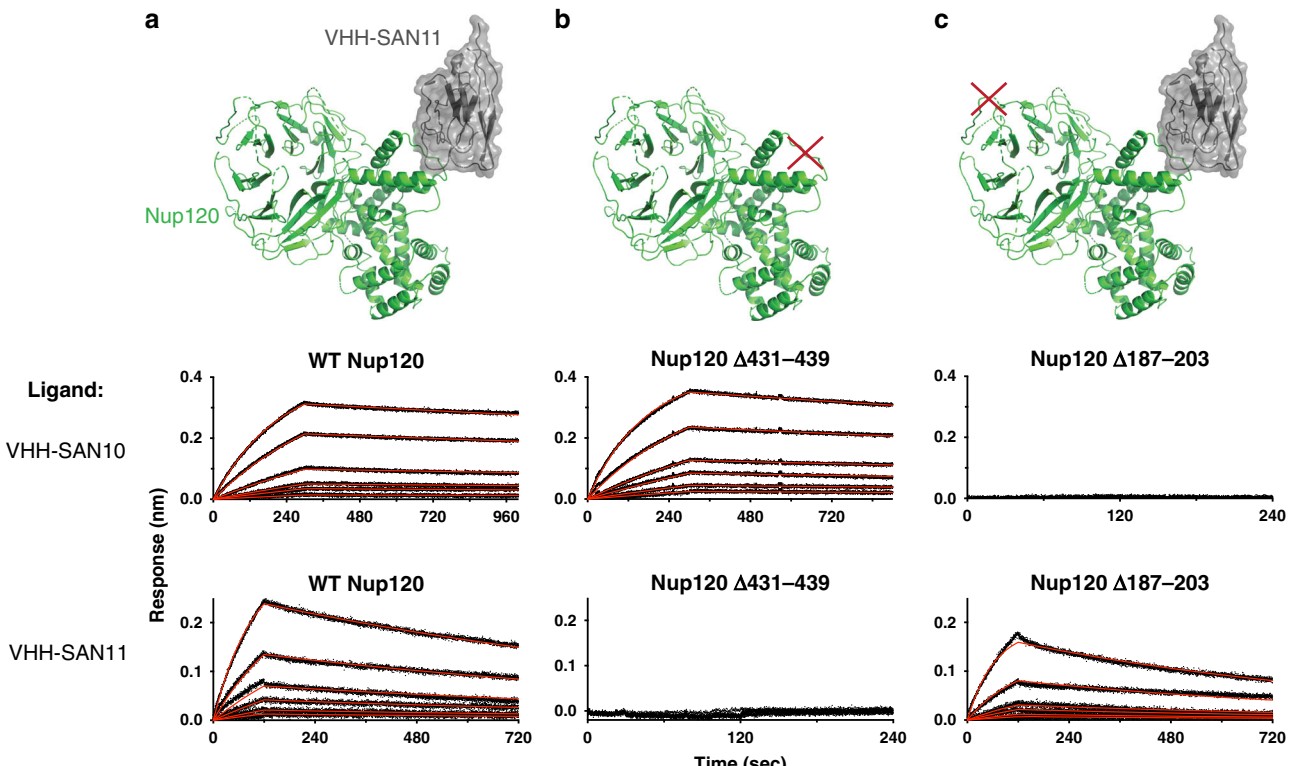

**Fig. 4 Mutational analysis of Nup120 confirms the binding site of VHH-SAN10.** Bio-layer interferometry (BLI) showing association and dissociation kinetics for VHH-SAN10 and VHH-SAN11 with Nup120 mutants. Nanobodies with a biotinylated C-terminal Avi-tag were fixed (ligand) and nups used as analytes. Curves were corrected for buffer background. Each set of curves is a twofold dilution series from 10 nM analyte. Data are indicated by the black dotted lines and the red lines show the globally fitted curves. **a** Structure of Nup120$_{1-757}$-VHH-SAN11 and BLI data of wild type (WT) Nup120$_{1-757}$ as the analyte. **b** Illustration and BLI data of Nup120$_{1-757}$ Δ431–439 (GSSx3) as the analyte. **c** Illustration and BLI data of Nup120$_{1-757}$ Δ187–203 (GGSx5) as the analyte.

production of each nanobody under control of the high expression *GAL* promoter in a wild type yeast strain. We observed no fitness defects upon nanobody expression at both 30 °C and 37 °C (Fig. 6a). Given the small size of a nanobody relative to the NPC, this was not entirely surprising, but also suggests that none of the surfaces occupied by the nanobodies are essential for NPC integrity, assuming that the nanobodies can find their targets in the cell. We therefore asked whether the nanobodies localized to the NPC and the NE in vivo, thus examining their ability to bind their targets within the context of the assembled NPC. To this end, we fused mKate2, a monomeric far-red fluorescent protein[40], to the C terminus of each nanobody. We then expressed these nanobody-mKate2 fusions in a yeast strain with endogenously GFP-tagged Nup120 (Nup120-GFP) as a Y complex and NPC marker, and examined co-localization. The expression of the nanobody-mKate2 fusions had only minor defects on fitness at 37 °C, with the exception of VHH-SAN7, which recognizes Sec13 (Fig. 6a). As none of the nanobody-mKate2 fusions were lethal, we converted the *GAL* promoter to the Nup120 promoter. We opted to perform the localization analysis using the weaker Nup120 promoter to maintain a similar stoichiometry of the nanobody to NPC components[28] rather than to vastly over-express the nanobody or have any confounding effects from GAL induction.

In the absence of any nanobody expression, we observed clear nuclear rim fluorescence for Nup120-GFP, while mKate2 alone produced diffuse fluorescence throughout the cytoplasm and the nucleus (Fig. 6b, 'Mock'). Of the three Nup85 nanobodies, one exhibited strong co-localization with Nup120 at the nuclear rim (VHH-SAN3) while the other two (VHH-SAN1 and 2) displayed

reduced rim colocalization. VHH-SAN4 and 5 nanobodies, which recognize Nup133 at its N-terminal β-propeller, did not colocalize with Nup120 but instead were distributed diffusely throughout the cell. VHH-SAN6, which recognizes Nup145C in the Y complex hub, also strongly enriched with Nup120 at the nuclear rim. VHH-SAN7 formed puncta in the cytoplasm, possibly due to binding Sec13 in the COPII vesicle coat or SEA complex, rather than forming a complex with the copies of Sec13 in the NPC[47–49]. We suggest that binding of VHH-SAN7 to these cytoplasmic copies of Sec13 may also account for the fitness defect of this strain. The majority of cells expressing both Nup84 nanobodies (VHH-SAN8 and 9), also showed strong nuclear rim localization with Nup120. However, VHH-SAN9 had a curious effect on Nup120 localization. While some cells expressing VHH-SAN9 showed strong colocalization with Nup120-GFP, many other cells had no or diffuse localization of Nup120-GFP on the NE to which VHH-SAN9 was localized. For the Nup120 nanobodies, both VHH-SAN11 and 12 showed strong localization at the nuclear rim. Finally, VHH-SAN12, which recognizes Nic96, presented as foci at the nuclear rim. Taken together, we find that the expression of our nanobodies in *S. cerevisiae* does not obviously affect cellular fitness. Many of these nanobodies localize to the NPC in vivo, confirming their potential as tools for future experiments in understanding NPC composition and assembly.

## Discussion

Here we describe a 12-member library composed of nanobodies that bind both the Y complex and Nic96. The nanobodies bind their targets tightly, with all but one nanobody having an

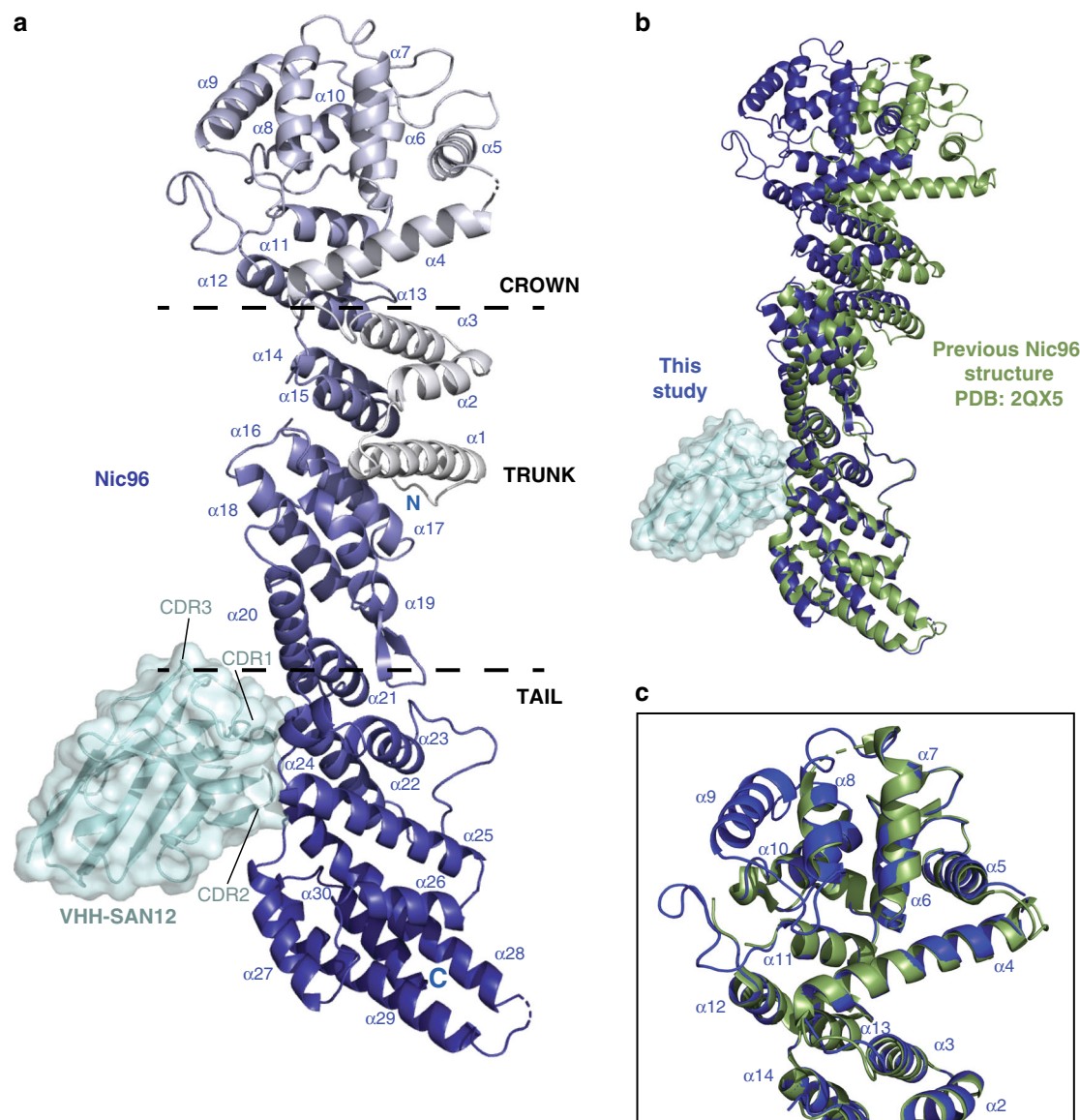

**Fig. 5 Structure of Nic96-VHH-SAN12 highlights the flexibility of the ACE1 fold. a** The structure of Nic96-VHH-SAN12. Nic96 is shown in gradient color from white to dark blue and VHH-SAN12 is shown in light blue. Helices are labeled on Nic96, along with the complementarity determining region (CDR) loops on VHH-SAN12. Boundaries of the ancestral cotamer element (ACE1) fold modules are indicated by the dashed lines. **b** Superposition of Nic96 with the previously solved structure (PDB: 2QX5[37]) (shown in green). The alignment was done only in the tail module of the protein. **c** Superposition of the Nic96 crown module between the two structures. This structure of Nic96, shown in dark blue and the previous structure, shown in green.

equilibrium binding constant of <10 nM. Through both SEC and X-ray crystallography, we mapped their general binding sites or epitopes. The recognized epitopes range from relatively unstructured loops to clefts between domain interfaces. We also expressed these nanobodies in yeast cells to assess their ability to localize to NPCs in their native cellular environment. Armed with this information, we can now interpret the heterogeneity of the observed in vivo effects and identify elements of this library that can aid in future NPC assembly studies.

The Nup85 crown is thought to be adjacent to the Nup82 complex, where both VHH-SAN2 and 3 bind[24,30] (Fig. 7a, b). This correlates well with the localization of VHH-SAN2 being largely diffuse throughout the cell, suggesting that the VHH-SAN2 binding site is occluded. However, there is still some enrichment at the NE with Nup120, suggesting the Nup82-Nup85 tether may be flexible or dynamic, allowing for a fraction of the nanobody pool to still bind. Another possibility is that some Nup85 epitopes may be accessible, while others are not. This

would suggest that Nup85 is differently assembled in different parts of the NPC. Both VHH-SAN1 and VHH-SAN3 are strongly enriched at the NE. However, the expression of VHH-SAN1 also yielded the unexpected formation of VHH-SAN1-Nup120 puncta away from the NE. We hypothesize that this nanobody may weaken the affinity of the Y complex to the NPC assembly. These puncta could represent Y complexes that are slower to incorporate into the NPC assembly or potentially dissociated Y complexes from assembled NPCs. In either case, this is only a modest disruption, as the cells have no obvious growth defect.

Both Nup133 nanobodies (VHH-SAN4 and 5) were unable to localize to the NE in yeast cells (Fig. 6b). These nanobodies are most likely blocked from binding in the assembly by the Arf-GAP1 lipid packing sensing (ALPS) motif[50] (Fig. 7b, d). Nup133 is thought to be anchored to the NE by its ALPS motif on its N-terminal β-propeller domain[51,52], which is on the same face of Nup133 as the epitopes for VHH-SAN4 and 5[20]. This suggests that the ALPS interaction with the membrane outcompetes the

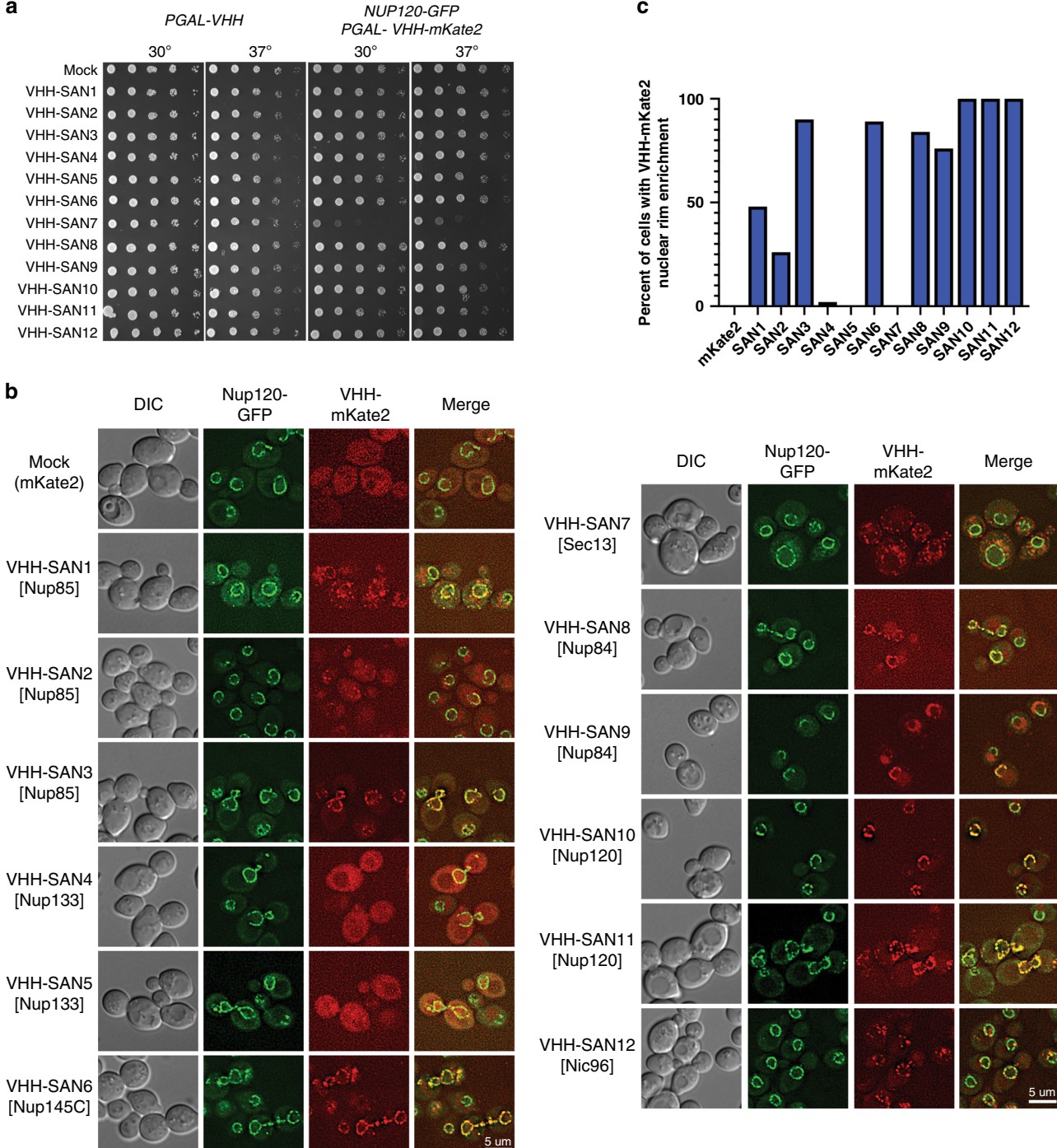

**Fig. 6 Expression of nanobody-mKate2 fusions in vivo. a** Spot assays for nanobody and nanobody-mKate2 expression. Strain construction is indicated at the top of the figure, along with assay temperature. Pictures were taken after 48 h on 2% galactose containing agar media. Mock specifies mKate2 expression alone. **b** Fluorescence images of cells expressing endogenously tagged Nup120-GFP and plasmid expressed *NUP120*-VHH-mKate2 in DIC, GFP, and far-red channels with a merged overlay. Target of each nanobody is listed in brackets. Scale bar: 5 μM. **c** Percent of cells with VHH-mKate2 exhibiting nuclear rim enrichment, *n* = 50 cells per strain over one experiment.

binding of the nanobodies. While VHH-SAN4 has both fast on and off kinetics and the weakest binding affinity (230 nM), VHH-SAN5 binds Nup133 tightly in vitro (10 nM). This implies that the membrane interaction of the ALPS motif must have an even higher affinity or the epitope for VHH-SAN4 and 5 on Nup133 is blocked by the membrane very soon after new copies of Nup133 are synthesized, since the nanobodies are constitutively expressed.

Similar to VHH-SAN1, the Nup145C-specific VHH-SAN6 redistributes some Nup120 (and potentially Y complex) into cytoplasmic puncta, but does so to a lesser extent, along with its colocalization on the NE (Fig. 6b). Like VHH-SAN1 expression, the presence of these puncta has no effect on the fitness of the strain, so the pool of Y complex present on the NE must still be sufficient for proper cellular function. We cannot exclude at present that the expression of nanobodies like VHH-SAN1 might

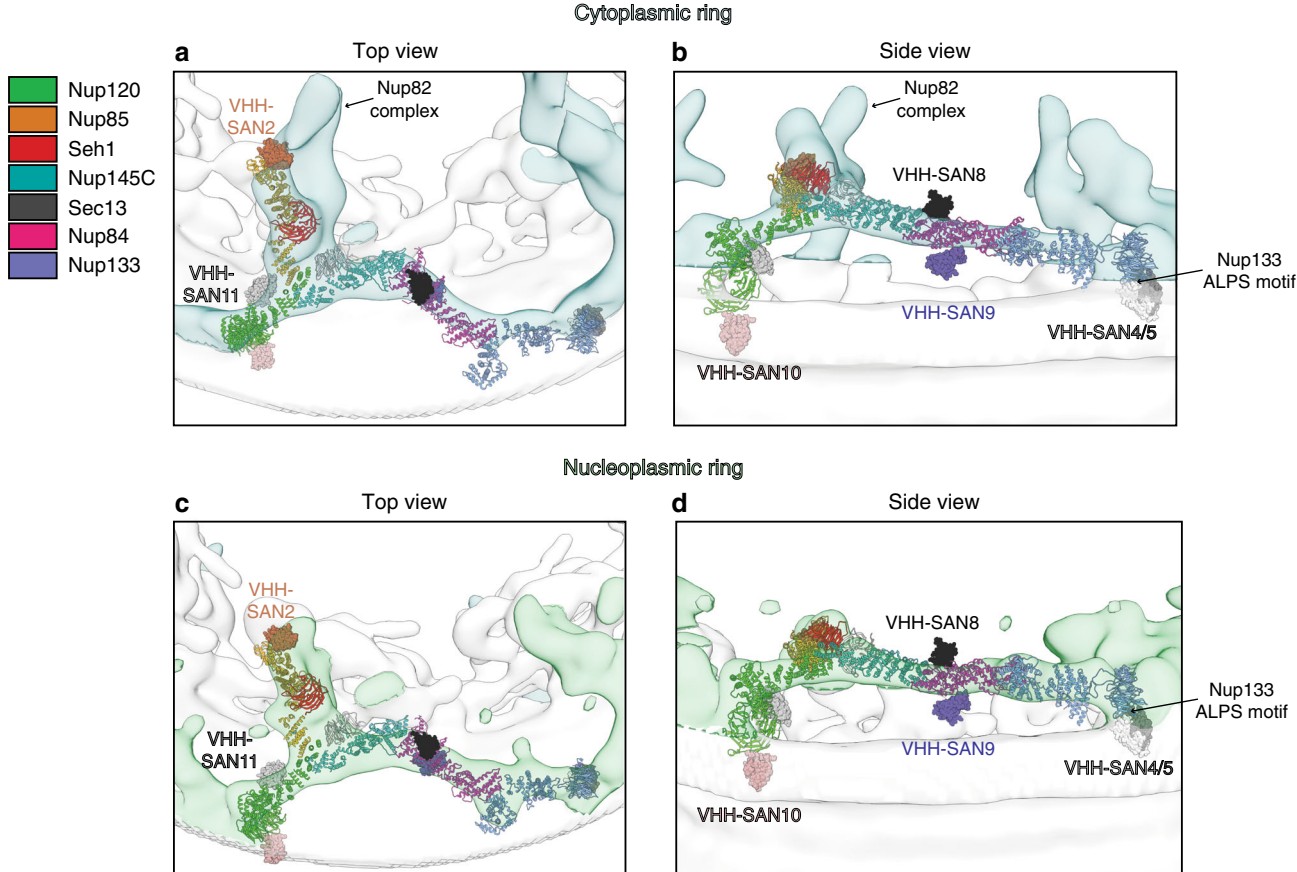

**Fig. 7 Docking of the Y complex and nanobodies into the cryo-ET map of the NPC. a** Top and **b** side view of the cryo-ET map from *S. cerevisiae* (EMD-10198) with the density for corresponding to the cytoplasmic ring of the NPC colored in light blue. A top-scoring global fit for the composite *S. cerevisiae* Y complex model shown in both a top and side view. Y complex nups colored as indicated in the legend (right). Nanobodies were not included during global fitting, nanobody-nup complexes were superposed after docking. Nanobodies are shown with surface rendering and labeled with matching colors. Density corresponding to the Nup82 complex is indicated. The position of the Nup133 ALPS motif adjacent to the nuclear envelope is also indicated. **c** Top and **d** side view cryo-ET map with the density corresponding to the nucleoplasmic ring of the NPC colored in light green. Top-scoring global fit for the composite *S. cerevisiae* Y complex model shown in both a top and side view. Nanobody positioning, coloring, and labeling as in **a**, **b**.

exert more subtle effects, for example by affecting the extent or rate of nuclear import/export of select cargoes. The Sec13 nanobody, VHH-SAN7, is the only nanobody in the library that decreases fitness as a mKate2 fusion, but not when expressed alone (Fig. 6a). From our fluorescence localization data, VHH-SAN7 exists mostly in cytoplasmic puncta, potentially due to the fact that Sec13 is present not only in the NPC, but also the COPII vesicle coat and SEA complex[47,48]. We hypothesize that VHH-SAN7 binds the copies of Sec13 outside of the NPC present in the cytoplasm, possibly by interacting with these copies of Sec13 more quickly after translation. Although we know VHH-SAN7 can bind Sec13 when assembled into the Y complex hub in vitro, it is also possible that its binding site is occluded in the context of the assembled NPC (Supplementary Fig. 1).

The Nup84 specific nanobodies (VHH-SAN8 and 9) behave differently due to binding opposite faces of Nup84. VHH-SAN8 binds the 'top surface of Nup84, away from the NE (Fig. 7b, d). This most likely explains why VHH-SAN8 colocalizes strongly with Nup120-GFP and causes no fitness defects. On the other hand, the expression of VHH-SAN9 has a peculiar effect on the localization of Nup120-GFP. Many cells showed diffuse fluorescence for VHH-SAN9, but in the general curvature of the NE. In many cases, we observed similar diffuse NE fluorescence of Nup120-GFP on part of the NE, but crisp NE rim fluorescence on the part of the nucleus where VHH-SAN9 was not present. VHH-

SAN9 binds the side of Nup84 adjacent to the NE (Fig. 7b, d). It is possible that the presence of VHH-SAN9 disrupts the Y complex assembly on the NE, prying the Y complex away from the membrane. Interestingly, this occurs on only some of the NPCs within the same cell. Further investigation into the state of these NPCs and the NE is ongoing.

Both Nup120 specific nanobodies colocalize well with Nup120-GFP. The expression of VHH-SAN10 has little effect on the distribution of rim fluorescence of Nup120-GFP. Residues 197–216 are hypothesized to also be an ALPS motif and the overlapping residues 187–203 are required for VHH-SAN10 binding[52]. Docking of the Y complex into the cryo-ET map of the scNPC suggests this face of Nup120 to be positioned adjacent to the membrane[24] (Fig. 7b, d). If Nup120 does bind the membrane at this loop, the nanobody likely outcompetes the affinity of Nup120 for the membrane. This would also suggest that the membrane-attachment by Nup120 may not be critical for NPC assembly and function. VHH-SAN11 had a more pronounced effect on the localization of Nup120-GFP. Some protrusions emanating from the NE were observed to have both Nup120 and VHH-SAN11, along with some foci in the cytoplasm.

VHH-SAN12 binds Nic96 between its trunk and tail modules. The interaction between the CDR loops of VHH-SAN12 and these trunk-tail interface helices maintains Nic96 in a different conformation than previously observed by X-ray crystallography[45,46].

Most of the VHH-SAN12-mKate2 fusion enriched at the NE along with Nup120-GFP. However, VHH-SAN12 formed foci, both on the NE and in the cytoplasm and within the nucleus, while Nup120-GFP showed an even distribution on the NE. There are two possible explanations that can account for these foci. Either VHH-SAN12-mKate2 expression was limiting or not every NPC displays Nic96 in a conformation or configuration accessible for VHH-SAN12 binding. The concept of NPC heterogeneity has been observed and discussed in the literature, both in terms of its composition and in size[53–55]. Whether this interesting fluorescence is indeed due to NPC heterogeneity within the same cell is also under ongoing investigation.

The nanobody suite describes here provides a set of tools for studying the NPC assembly both in vitro and in vivo. In vitro, the nanobodies bind tightly to their targets and have enabled structural analysis of multiple nups at higher resolution than reported earlier or that evaded previous attempts by X-ray crystallography altogether[20]. In vivo, many nanobodies colocalize with the Y complex and therefore can be used as cellular tools in future studies on the NPC. For example, the nanobody library could enable subunit identification in future cryo-ET studies. In several cases, such as VHH-SAN9 and VHH-SAN12, they have also provided questions for further study on NPC assembly and heterogeneity. Along with ref. [20], we also detailed the library's breadth in antigen recognition, owing to its wide diversity in both CDR sequence and length. Overall, our data highlight the exciting potential of this nanobody library to be used as tools for both in vitro and in vivo studies of the NPC and pave the way for future explorations of NPC assembly and composition in *S. cerevisiae*.

## Methods

**Construct generation**. All Nups (Nup120, Nup85, Seh1, Nup145C, Sec13, Nup84, Nup133, Nic96) were cloned from *S. cerevisiae* and expressed recombinantly in *E. coli*. Expression constructs for His-tagged Nup84, Nup85, Nup85$_{200–383}$ (Nup85$_{crown}$), Nup85$_{1–564}$-Seh1 (Nup85$_{crown-trunk}$-Seh1), Nup120$_{855–1037}$-Nup145C$_{34–712}$-Sec13, Nup120$_{1–757}$, Nup120-Nup145C-Sec13, Sec13-Nup145C$_{320–411}$ (Sec13-Nup145C$_{blade}$), and Nic96$_{186–839}$ were published previously[18,35,40,45,56]. In addition, we used N-terminally fused 14xHis, bdSUMO tagged full-length Nup133 and N-terminally fused 6xHis tagged Nup133$_{55–481}$ (Nup133$_{NTD}$).

Upon VHH selection by phage display and ELISA, each VHH was sub-cloned for expression. Each VHH sequence was N-terminally fused with a 14xHis bdSUMO tag[57] and cloned into a T7-promoter-based bacterial expression vector with ampicillin resistance. A separate construct for each VHH was created with a C-terminally fused Avi-tag[36] for biotinylation. Primers used for cloning are listed in Supplementary Table 1.

**Protein expression and purification**. Nup120-Nup145C$_{34–712}$-Sec13, Nup85-Seh1, Nup120$_{855–1037}$-Nup145C$_{34–712}$-Sec13, Nup84, Nic96$_{189–839}$, Nup133$_{NTD}$, Sec13-Nup145C$_{blade}$, Nup85$_{crown-trunk}$-Seh1, and Nup85$_{crown}$ expression vectors were transformed into *E. coli* LOBSTR-RIL(DE3)[58] (Kerafast) cells and protein production was induced with 0.2 mM IPTG at 18 °C for 12–14 h. Cells were collected by centrifugation at $6000 \times g$, resuspended in lysis buffer (50 mM potassium phosphate pH 8.0, 500 mM NaCl, 30–40 mM imidazole, 3 mM β-mercaptoethanol (βME), 1 mM PMSF) and lysed using a high-pressure cell homogenizer (Microfluidics LM20). The lysate was cleared by centrifugation at $12,500 \times g$ for 25 min. The soluble fraction was incubated with Ni Sepharose 6 Fast Flow beads (GE Healthcare) for 30 min on ice. After washing the beads with lysis buffer, the protein was eluted (250 mM imidazole pH 8.0, 150 mM NaCl, 3 mM βME).

All VHH constructs and Nup133 were transformed, grown, harvested, and lysed as above. For Avi-tagged VHH constructs, 20 mM biotin was added to the cultures prior to IPTG induction. After lysis, the soluble fraction was incubated with Ni Sepharose 6 Fast Flow beads (GE Healthcare) for 30 min on ice. The beads were then washed with lysis buffer and transferred to low imidazole buffer (50 mM potassium phosphate pH 8.0, 500 mM NaCl, 10 mM imidazole, 3 mM βME) along with 10 µg SEN-p protease and incubated 2 h at 4 °C. The flow through containing the cleaved protein was collected, along with a 2-column volume (CV) wash with low imidazole buffer. Cut tags and uncut protein was eluted as above.

Buffers used in further purification are as follows: gelfiltration (GF) buffer (150 mM NaCl, 10 mM Tris/HCl pH 7.5, 1 mM DTT, 0.1 mM EDTA), Dialysis buffer (100 mM NaCl, 10 mM Tris/HCl pH 7.5, 1 mM DTT, 0.1 mM EDTA), S buffers (0 or 1 M NaCl, 10 mM Tris/HCl pH 7.5, 1 mM DTT, 0.1 mM EDTA) and Q buffers (0 or 1 M NaCl, 20 mM HEPES/NaOH pH 8.0, 1 mM DTT, 0.1 mM EDTA).

After Ni purification, Nup120-Nup145C$_{34–712}$-Sec13, Nup120$_{855–1037}$-Nup145C$_{34–712}$-Sec13, Nup85-Seh1, Nup84, Nic96$_{189–839}$, Nup133$_{NTD}$, Sec13-Nup145C$_{blade}$, Nup85$_{crown-trunk}$-Seh1, Nup120$_{1–757}$, and Nup133 were incubated with 3C protease and dialyzed into dialysis buffer overnight. Nup120-Nup145C$_{34–712}$-Sec13, Nup84, Nup85$_{crown}$, and Nup120$_{1–757}$ were run over a HiTrap SP FF column (GE Healthcare), collecting the flow through. Nup133 was loaded onto a MonoQ column (GE Healthcare) and eluted over a gradient of 100–700 mM NaCl, collecting the peak. Nup85$_{crown-trunk}$-Seh1, Nup120-Nup145C$_{34–712}$-Sec13, Nup85-Seh1, Nup84, Nic96$_{189–839}$, and Nup133 were concentrated and loaded onto a pre-equilibrated Superdex S200 16/60 column (GE Healthcare) in GF buffer. Nup133$_{NTD}$, Sec13-Nup145C$_{blade}$, Nup85$_{crown}$, and every VHH were concentrated and loaded onto a pre-equilibrated Superdex S75 16/60 column (GE Healthcare) in GF buffer. All fractions were analyzed by SDS-PAGE, peaks were pooled and concentrated.

For all Nup-VHH complexes, the proteins were incubated for 30 min on ice. The incubated mixtures were then run over a Superdex S75 or S200 10/300 column pre-equilibrated in GF buffer. Fractions containing the complexes were pooled and concentrated. To assemble the Y complex, we first mix 1.5× molar excess of Nup85-Seh1 with Nup120-Nup145C$_{34–712}$-Sec13. After incubation for 30 min on ice, the complex was run over a Superdex S200 10/300 column in GF buffer. Fractions containing all five proteins were pooled and concentrated. We also mixed 1.5× molar excess of Nup84 with Nup133 and followed the same protocol. These two complexes were then mixed together with 1.5× molar excess of Nup84-Nup133 and run over a Sepharose 6 10/300 column (GE Healthcare). Fractions containing all Y complex components were pooled and concentrated.

**VHH library and M13 phage generation**. Alpaca immunization and library generation were done as previously described[59]. The animal was purchased locally, maintained in the pasture, and immunized following a protocol authorized by the Tufts University Cummings Veterinary School Institutional Animal Care and Use Committee (IACUC). The animal was immunized against recombinantly expressed full-length Y complex (Nup120-Nup85-Seh1-Nup145C-Sec13-Nup84-Nup133). The library was then grown to mid-log phase in 100 ml SOC with 50 µg/ml ampicillin. Then, the culture was infected with 100 µl 10$^{14}$ PFU/ml VCSM13 helper phage. Following 2 h incubation at 37 °C, the cells were harvested by centrifugation and re-suspended in 100 ml 2YT, 0.1% glucose, 50 µg/ml kanamycin, and 50 µg/ml ampicillin. Cultures were incubated overnight at 30 °C, then centrifuged for 20 min at $7700 \times g$, followed by phage precipitation from the resulting supernatant with 1% PEG-6000, 500 mM NaCl at 4 °C, and resuspended in PBS.

**Selection of VHHs by phage display**. VHHs were selected by panning against Nup120$_{855–1037}$-Nup145C$_{34–712}$-Sec13, Nup85-Seh1, Nic96$_{189–839}$, Nup84, and Nup133$_{55–481}$. Hundred micrograms of recombinant protein was biotinylated by coupling Chromalink NHS-biotin reagent (Solulink) to primary amines for 90 min in 100 mM sodium phosphate pH 7.4, 150 mM NaCl. Unreacted biotin was removed using a Zeba desalting column (Thermo Fisher). Biotin incorporation was monitored using absorbance at 354 nM. 100 µl MyOne Streptavidin-T1 Dynabeads (Life Technologies) were blocked in 2% (w/v) bovine serum albumin (Sigma) in PBS for 2 h at 37 °C. Twenty micrograms of biotinlyated antigen in PBS was added to the blocked beads and incubated for 30 min at 25 °C with agitation. The beads were then washed three times in PBS and 200 µl of 10$^{14}$ PFU/ml M13 phage displaying the VHH library were added in 2% BSA in PBS for 1 h at room temperature. The beads were then washed 15 times with PBS, 0.1% Tween-20 (PBST). Phage was eluted by the addition of *E. coli* ER2738 (NEB) for 15 min at 37 °C, followed by elution with 200 mM glycine, pH 2.2, for 10 min at 25 °C. The eluate was neutralized with 1 M Tris/HCl pH 9.1, pooled with the *E. coli* culture, and plated onto 2YT agar plates supplemented with 2% glucose, 5 µg/ml tetracycline, and 10 µg/ml ampicillin, and grown overnight at 37 °C. A second round of panning was performed with the following modifications: 2 µg of biotinylated antigen was used as bait, and incubated with 2 µl 10$^{14}$ PFU/ml M13 phage displaying the first-round VHH library for 15 min at 37 °C, followed by 15 washes in PBST.

**ELISA**. Following two rounds of phage panning, 96 colonies were isolated in 96-well round-bottom plates and grown to mid-log phase at 37 °C in 200 µl 2YT, 10 µg/ml ampicillin, 5 µg/ml tetracycline, induced with 3 mM IPTG and grown overnight at 30 °C. Plates were centrifuged at $12,000 \times g$ for 10 min, and 100 µl of supernatant was mixed with an equal volume of 5% (w/v) nonfat dry milk in PBS. This mixture was added to an ELISA plate coated with 1 µg/ml antigen. Following four washes with 1% Tween-20 in PBS, anti-llama-HRP antibody (Bethyl) was added at a 1:10,000 dilution in 5% (w/v) nonfat dry milk in PBS for 1 h at 25 °C. The plate was developed with fast kinetic TMB (Sigma) and quenched with 1 M HCl. Absorbance at 450 nm was determined in a plate reader (Spectramax; Molecular Devices).

**Biolayer interferometry**. Streptavidin biosensor tips were pre-incubated in BLI buffer (10 mM Tris/HCl pH 7.5, 150 mM NaCl, 1 mM DTT, 0.1 mM EDTA, 0.05% Tween-20, 0.1% bovine serum albumin) for 10 min, followed by immobilization of

biotinylated, C-terminally Avi-tagged nanobody ligands to between 0.2–0.5 nm over 40–60 s. After dipping the coated biosensor tip in BLI buffer for 1 min, association was measured in analyte over 1–80 min. Dissociation was measured in BLI buffer for 1–220 min. All binding sensorgrams were recorded on a forteBIO OctetRED96 instrument. All fits were done using global, 1:1 kinetic binding parameters using the Octet data analysis software.

**Protein crystallization**. Initial hits of Nic96$_{186-839}$-VHH-SAN12 were obtained at 18 °C in 1 day in a 96-well sitting drop tray with a reservoir containing 8% (w/v) PEG 8,000 and 0.1 M tri-sodium citrate pH 5.0 (Protein Complex suite, Qiagen). Hanging drops of 1 µl protein at 6 mg/ml and 1 µl of precipitant (7–10% (w/v) PEG 8000 and 0.1 M tri-sodium citrate pH 5.0) incubated at 18 °C produced diffraction quality rod-shaped crystals in 3 days. Crystals were transferred into a cryo-protectant solution containing the crystallization condition with 15% (v/v) glycerol and cryo-cooled in liquid nitrogen.

Initial hits of Nup120$_{1-757}$-VHH-SAN10/11 were obtained at 18 °C in 3 days in a 96-well sitting drop tray with a reservoir containing 20% (w/v) PEG 8000, 0.2 M magnesium chloride, 0.1 M Tris/HCl pH 8.5 (JCSG + suite, Qiagen). Hanging drops of 1 µl protein at 2 mg/ml and 1 µl of precipitant (19% (w/v) PEG 8000, 0.1 M magnesium chloride, 0.1 M Tris/HCl pH 8.5) incubated at 18 °C yielded large rod-shaped crystals in 4 days. Crystals were transferred into a cryo-protectant solution containing the crystallization condition with 20% (v/v) glycerol and cryo-cooled in liquid nitrogen.

Initial hits of Nup85$_{1-564}$-Seh1-VHH-SAN2 were obtained at 18 °C in 1 day in a 96-well sitting drop tray with a reservoir containing 1 M ammonium sulfate and 0.1 M sodium acetate pH 5.0 (Protein Complex suite, Qiagen). Crystals significantly improved by changing the buffer to di-sodium succinate pH 5.5 and the addition of 4% (v/v) 1-propanol. Sitting drops of 0.2 µl protein at 17 mg/ml and 0.2 µl of precipitant yielded diffraction-quality, hexagonal rod-shaped crystals in 3 days that continued to grow over 3 weeks at 18 °C. Crystals were transferred into a cryo-protectant solution containing the crystallization condition with 15% (v/v) PEG 200 and cryo-cooled in liquid nitrogen.

**Structure determination**. Data collection was performed at the advanced photon source end station 24-IDC. All data processing steps were carried out with programs provided through SBgrid[60]. Data reduction was performed using HKL2000[61]. Statistical parameters of data collection and refinement are all given in Table 2. All manual model building steps were carried out with Coot[62] and phenix.refine[63] was used for iterative refinement. Structure figures were created in PyMOL (Schrödinger LLC).

The structure of Nic96$_{186-839}$-VHH-SAN12 was solved by MR using Phaser-MR in PHENIX. A two-part MR solution was obtained by sequentially searching with models of Nic96 and VHH-SAN12. For Nic96, we used the previously solved structure from *S. cerevisiae* (PDB:2QX5)[45]. For VHH-SAN12, we used a nanobody structure with its CDR loops removed (PDB:1BZQ)[64]. The asymmetric unit contains one copy of Nic96$_{186-839}$-VHH-SAN12. Near the end of refinement, TLS refinement was used, with a significant impact on lowering the R factors.

The structure of Nup120$_{1-757}$-VHH-SAN10/11 was solved by MR using Phaser-MR in PHENIX, just as Nic96$_{186-839}$-VHH-SAN12. We used the previously solved *S. cerevisiae* Nup120$_{1-757}$ structure as the search model (PDB:3HXR)[40]. The asymmetric unit contains one copy of the complex. Many sidechains were removed from the model due to poor density. Secondary structure restrains were used throughout refinement. Regarding the two nanobodies, density only for VHH-SAN11 was present on the map.

The structure of Nup85$_{crown-trunk}$-Seh1-VHH-SAN2 was solved by MR using Phaser-MR in PHENIX. A solution was found by searching with the previously solved *S. cerevisiae* Nup85$_{1-564}$-Seh1 structure as the model (PDB:3EWE)[18]. The asymmetric unit contains one copy of the complex. Large difference density was present near the crown domain of Nup85, the known binding site of VHH-SAN2 biochemically (Supplementary Fig. 4), and we were able to manually place a model after refinement of Nup85$_{crown-trunk}$-Seh1. Since the CDR loops provide a significant amount of density nearest to Nup85, a model was generated by SWISS-MODEL. Near the end of refinement, TLS parameters were used. For the final structure, occupancies for the VHH-SAN2 residues outside of the 2Fo-Fc density at 1σ were set to zero.

**Yeast strain construction**. The Nup120-GFP strain was made as previously described[40]. In brief, C-terminal GFP-tagging was achieved by homologous recombination in a BY4741 background, using pFA6a-GFP(S65T)-kanMX6 as a template for C-terminal modifications[65]. Strains were selected on G418 plates (200 µg/ml) and verified by PCR. Nanobody expression for fluorescence experiments was done on a plasmid based on pAG416GAL-ccdB (Addgene). The plasmid was modified with the Nup120 promoter (500 bp upstream of the ORF) in place of the GAL promoter and each nanobody followed by a fused C-terminal mKate2 tag. Strains were selected and maintained by growth on SD-Ura media and verified by sequencing. Nanobody expression for toxicity experiments was done on a plasmid based on pAG416GAL-ccdB (Addgene). Each VHH sequence was inserted downstream of the GAL promoter and strains were selected and maintained by growth on SC-Ura media.

### Table 2 Data collection and refinement statistics.

| Protein | Nic96$_{189-839}$, VHH-SAN12 | Nup120$_{1-757}$, VHH-SAN10, VHH-SAN11 | Nup85$_{1-564}$-Seh1, VHH-SAN2 |
|---|---|---|---|
| PDB code | 6X07 | 6X06 | 6X08 |
| Organism | *S. cerevisiae, V. pacos* | | |
| Data collection | | | |
| Space group | P 2$_1$ 2$_1$ 2$_1$ | P 6$_1$ | P 6$_4$ 2 2 |
| Cell dimensions | | | |
| a, b, c (Å) | 48.1, 79.5, 283.3 | 193.2, 193.2, 78.1 | 234.0, 234.0, 139.4 |
| α, β, γ (°) | 90, 90, 90 | 90, 90, 120 | 90, 90, 120 |
| Resolution (Å) | 60.9-2.1 (2.2-2.1)$^a$ | 83.7-4.3 (4.4-4.3) | 117.0-4.2 (4.4-4.2) |
| $R_{p.i.m.}$ (%) | 3.8 (67.7) | 5.2 (51.2) | 5.6 (66.5) |
| $I/σ$ | 23.9 (1.2) | 18.7 (1.4) | 19.1 (1.3) |
| CC$_{1/2}$ | 1 (0.69) | 0.94 (0.52) | 0.98 (0.64) |
| | 99.4 (98.9) | 99.5 (99.2) | 99.9 (99.9) |
| Completeness (%) | | | |
| Redundancy | 7.2 (6.9) | 7.3 (7.2) | 37.9 (39.9) |
| Refinement | | | |
| Resolution range (Å) | 60.9-2.1 | 83.7-4.3 | 117.0-4.2 |
| No. reflections | 64,310 | 11,641 | 16,995 |
| $R_{Work}/R_{Free}$ | 21.6/25.8 | 33.6/35.9 | 31.3/34.7 |
| No. atoms | | | |
| Protein | 5,786 | 3818 | 6890 |
| Water | 140 | 0 | 0 |
| B factors (Å$^2$) | | | |
| Protein | 88.6 | 241.0 | 207.3 |
| Water | 69.4 | – | – |
| r.m.s. deviations | | | |
| Bond length (Å) | 0.011 | 0.002 | 0.006 |
| Bond angles (°) | 1.101 | 0.605 | 0.949 |

$^a$Values in parenthesis are for the highest-resolution shell (10% of the data). One crystal was used for each dataset.

**Fluorescence microscopy**. Strains were grown overnight in SD-Ura media (CSM-Ura (Sunrise Science), Yeast nitrogen base with ammonium sulfate, 2% (v/v) glucose) at 30 °C, followed by 20-fold dilution into fresh SC-Ura media. After growth for 4–5 h at 30 °C to OD600 ~0.5, cells applied to a thin SD-Ura agar pad on a standard microscopy slide and imaged live on a DeltaVision Elite Widefield Deconvolution Microscope (GE Healthcare) using a 100× oil immersion objective with an sCMOS camera (Teledyne Photometics). Images were processed and analyzed in ImageJ[66].

**Reporting summary**. Further information on research design is available in the Nature Research Reporting Summary linked to this article.

## Data availability
Coordinates and structure factors have been deposited in the Protein Data Bank under PDB accession codes 6X06 (Nup120$_{1-757}$-VHH-SAN10/11), 6X07 (Nic96$_{186-839}$-VHH-SAN12), 6X08 (Nup85$_{1-564}$-Seh1-VHH-SAN2). The cryo-ET map used for docking of the Y complex and nanobodies is described elsewhere[24] and available from the Electron Microscopy Data Bank (EMDB) under accession number EMD-10198.

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

## Acknowledgements

We thank Nina C. Leksa and Xuanzong Guo for their initial work on the Nup120 structure and members of the Schwartz lab for discussions on the experiments and manuscript. We thank Karsten Weis for advice on fluorescence microscopy and experiments in yeast. The research was supported by the US NIH under grant number R01GM77537 (K.E.K., K.A., and T.U.S.) and T32GM007287 (S.A.N.). J.R.I. and H.P. were supported by an NIH Pioneer award. The X-ray crystallography work was conducted at the APS NE-CAT beamlines, which are supported by NIH award GM103403. Use of the APS is supported by the US Department of Energy, Office of Basic Energy Sciences, under contract no. DE-AC02-06CH11357.

## Author contributions

S.A.N. and T.U.S. designed the study. S.A.N. performed the experiments. K.A. solved the structure of Nic96$_{186-839}$-VHH-SAN12 and K.E.K. solved the structure of Nup120$_{1-757}$-VHH-SAN10/11. J.R.I. and H.P. conducted the alpaca immunization experiments and generation of the phagemid library. S.A.N. and T.U.S. interpreted the results and wrote the manuscript with input from K.A., K.E.K., and H.P.

## Competing interests

The authors declare no competing interests.
