## [Peer Review File · Nature Communications]

REVIEWER COMMENTS

Reviewer #1 (Remarks to the Author):

I recommend to reject this second manuscript, but more data on Nup85, Nup120, or Nup133 and the nanobodies binding to these proteins that are relevant to the biology of the NPC might justify the submission of high impact work by the same authors.

See attach for my analysis.

Jan Steyaert

VIB-VUB center for Structural Biology

Brussels

Reviewer #2 (Remarks to the Author):

This paper by the Schwartz laboratory reports the generation and validation of a library of nanobodies that target two key subcomplexes of the *S. cerevisiae* nuclear pore complex (NPC). Biochemical experiments establish that the obtained nanobodies bind to their nucleoporin targets with nanomolar affinity and accordingly yield stable and stoichiometric complexes in size-exclusion chromatography, which is the gold standard for protein-protein interaction analysis. To establish the molecular bases of nanobody binding to the nucleoporin targets, the authors determined a series of crystal structures. Finally, the authors established which of the nanobodies bind to their nucleoporin targets *in vivo* by expressing fluorescently tagged nanobodies in *S. cerevisiae*. Overall, this new nanobody library is a spectacular resource for the entire nucleocytoplasmic transport field and lays the groundwork for detailed studies of the conformational plasticity of NPCs during nucleocytoplasmic transport or for trapping NPC assembly intermediates. The authors report initial results in an accompanying paper that beautifully illustrate the potential of these nanobodies for such experiments. Generally, this is a very nice paper with a lot of high-quality data that represent a timely and important addition to the nucleocytoplasmic transport field. Unquestionably, the paper is of great interest to the broad readership of Nature Communications and I enthusiastically

recommend its publication without delay. I found only a few minor issues that the authors may want to consider prior to publication.

Minor points:

(1) Line 43. Generally, the paper appropriately cites the relevant literature. However, in discussing the attachment of the Nsp1•Nup49•Nup57 complex to the NPC via an interaction with Nic96, the authors should also cite the following three papers:

Chug H, Trakhanov S, Hülsmann BB, Pleiner T, Görlich D. (2015). Crystal structure of the metazoan Nup62•Nup58•Nup54 nucleoporin complex. *Science* 350, 106-110.

Stuwe T, Bley CJ, Thierbach K, Petrovic S, Schilbach S, Mayo DJ, Perriches T, Rundlet EJ, Jeon YE, Collins LN, Huber FM, Lin DH, Paduch M, Koide A, Lu V, Fischer J, Hurt E, Koide S, Kossiakoff AA, Hoelz A. (2015). Architecture of the fungal nuclear pore inner ring complex. *Science* 350, 56-64.

Fisher J, Teimer R, Amlacher S, Kunze R, Hurt E. (2015). Linker nups connect the nuclear pore complex inner ring with the outer ring and transport channel. *Nat. Struct. Mol. Biol.* 22, 774-81.

(2) Crystallographic analysis. The Rfree values of the Nup120(1-754)•VHH-SAN10•VHHSAN11 and of the Nup85(1-564)•Seh1•VHH-SAN2 crystal structures are somewhat high (35.9% and 34.7%, respectively). These high Rfree values may of course be a consequence of the low resolutions of these structures (4.3Å and 4.2Å, respectively). However, given that these nucleoporin structures have been previously solved at much higher resolution, one would expect much lower Rfree values. It may be worthwhile for the senior author to take another look at the space group assignments. I suspect that subgroups of the assigned high symmetry hexagonal space groups would actually yield lower Rfree values. I suspect the 6-fold axis may actually be imperfect and breaks down to a 2-fold or 3-fold axis. Regardless of this minor issue, both crystal structures clearly pinpoint the nanobody binding site on the target nucleoporin surface. Likewise, the evidence in Fig.4 and Supplementary Fig.5 supports the conclusion that VHH-SAN10 cannot be resolved because it binds to an unstructured loop of Nup120(1-754).

(3) Table 2. The authors are encouraged to report Ramachandran and rotamer geometry statistics, as well as ClashScore and MolProbity scores.

(4) Line 203. The conclusion that “nanobody binding causes a change in conformation” in Nic96 compared to previously solved crystal structures could be made more precise by specifying that the conformational change could also be induced by the novel, nanobody-mediated crystal lattice.

We thank the reviewers for their comments and for carefully reading the manuscript(s). Here is our point-by-point response. Response italicized for clarity.

Comments to Reviewer #1:

I have read two papers of the two back-to-back papers of the Schwartz lab with great interest... However, I have my doubts about the accompanying paper entitled 'A nanobody suite for yeast scaffold nucleoporins 1 provides details of the Nuclear Pore Complex structure'. Although the data are technically sound, the whole manuscript reads like an 'ennobled' materials and methods section with entire result sections and figures that normally fit the materials and methods section and the supplementary figures of recent high impact papers:

We respectfully disagree with the reviewer's view that this nanobody toolkit amounts to an 'ennobled' M&M section. We are not aware of a paper that listed this number of nanobodies, with this kind of in-depth biochemical AND cell biological AND structural analysis just in M&M. For reference, the most substantive nanobody-centered study on the nuclear pore complex that we are aware of, was published by the Görlich lab, DOI: 10.7554/eLife.11349. It characterized five nanobodies, contained no biochemical epitope mapping, structurally characterized just one nanobody and contained no live-cell data.

- Result section 'A nanobody library to the Y complex and Nic96' describes the selection of Nanobodies, following standard procedures

In our view, appropriate to show this as a result.

- Result section 'Nanobodies bind with varying kinetics, but strong affinities', describes the binding kinetics of these binders, measured by standard procedures

Showing real-time binding data for a toolkit of nanobodies seems appropriate to us and a perfectly reasonable result to illustrate. Many papers describing nanobodies do not contain such data, even though it is highly informative, probably because it is not trivial to generate.

- 'Figure 2. Bio-layer interferometry of nanobody-nup binding' just shows binding isotherms of the different Nanobodies to their different antigens?

Correct. We are happy to move this into the supplement if desired. But since the careful characterization of these twelve nanobodies is the main purpose of this paper, we believe that it is appropriate to show this data as a main figure.

- Figure 3 is a high resolution repeat of the right part of figure 1, panel B. More annoying is that the most interesting parts (the ones that are labelled with an Asterix) are the subject of the accompanying paper, proving again that the second manuscript corresponds to the M&M section of the first paper.

Figure 1b is a conceptual cartoon, Figure 3 is a composite structure. Figure 1b indicates all 12 nanobodies, Figure 3 only shows the subset that was co-crystallized. We believe that both figures make different points and are therefore individually appropriate.

- Figure 4 is another example that would be contained in the supplementary figures of a manuscript focussing on the biology of Nup120

Figure 4 shows an example of how we biochemically mapped the binding site of nanobodies, in light of moderate, and on-its-own not sufficient structural data. We could move it to the supplement, but since the paper is about the careful characterization of a toolkit of nanobodies we believe that it is informative and appropriate to have it as a main figure.

- Results section 'Several NPC nanobodies localize to the nuclear envelope *in vivo*' would correspond to the materials and method section on a future high impact paper on cryo-ET studies that makes use of nanobodies to enable subunit identification

To our knowledge, in vivo binding of nanobodies within a yeast cell is by no means an established fact, certainly not for the NPC. Particularly for cell biologists, who will hopefully make use of these reagents in the future, we believe that Figure 6 is likely the most important figure of the entire paper. For some labs, cryo-ET may be a way to employ these reagents, but this is only one possible application among many others, as Dr Steyaert would know better than probably anyone.

In conclusion, this manuscript describes the technical characterization of a number of valuable Nanobodies that bind the NPC. Several of them constitute 'a toolbox' and have great potential to be used for the integrated structural analysis of the Nuclear Pore Complex. Accordingly, the results on two Nanobodies that successfully served this purpose were lifted out of this manuscript and our the subject of the first manuscript.

However, the discussion of the second paper on the remaining Nanobodies is exceedingly hypothetical in its biological interpretation. The short version of this discussion reads like this: '... suggesting ... suggesting ... may be ... another possibility ... may be ... would suggest ... We hypothesize ... could represent ... or potentially ... most likely... This suggests ... We cannot exclude at present ... might exert ... potentially due ... We hypothesize ... possibly ... it is also possible ... This most likely ... Further investigation ... are hypothesized ... suggests ... likely ... may not be critical ... There are two possible explanations ... Whether ... is also under ongoing investigation'.

In light of the toolbox of nanobodies we generated, we thought that a somewhat open-ended discussion is called for. It surely does not read like any other discussion, but we found it in its form to be appropriate and we have, indeed, thought about it.

The authors will agree that many of the conclusions obtained on Nup85, Nup120, and Nup133 and the nanobodies binding to these proteins are premature and should be published

separately when more data become available. The scientists responsible for the generation and characterization of these valuable research tools should be authors on the first manuscript and included as co-authors on future manuscripts. But with the current data, this feels like two high impact papers for the price of one ...

If however two manuscripts are accepted as accompanying papers, all materials & methods relating to library constructions, phage display and selection should be brought together in the paper focussing on the description of the toolbox. In the current versions, the 'toolbox' paper refers to the 'structure paper' for immunizations, pannings, ELISAs, ...?

We have redistributed the methods between the two papers to have the detailed methods describing the phage display and selection, etc. in this paper.

Minor points:

- The letters C and T (Crown and Tail) are not explained in the legend of figure 1, panel B.

Good point. Legend adjusted

- The interpretation of figure 2 would largely increase if the graphical representations of figure 1, panel B (central part) are reiterated in figure 2.

We have modified Figure 2 accordingly

- Line 187: SAN10 and 11 add to a growing list of nanobodies that bind a variety of differently structured epitopes. It is not entirely clear to me what 'a variety of differently structured epitopes' means? We all agree that the nanobody binds one unique (structural) conformation of the epitope?

This is another good point, we should have worded this passage better. We wanted to point to the fact that nanobodies SAN10 and -11 bind in highly flexible regions, rather than to structured domains. We have modified the text.

Comments to Reviewer #2:

(1) Line 43. Generally, the paper appropriately cites the relevant literature. However, in discussing the attachment of the Nsp1•Nup49•Nup57 complex to the NPC via an interaction with Nic96, the authors should also cite the following three papers:

Chug H, Trakhanov S, Hülsmann BB, Pleiner T, Görlich D. (2015). Crystal structure of the metazoan Nup62•Nup58•Nup54 nucleoporin complex. *Science* 350, 106-110.

Stuwe T, Bley CJ, Thierbach K, Petrovic S, Schilbach S, Mayo DJ, Perriches T, Rundlet EJ, Jeon YE, Collins LN, Huber FM, Lin DH, Paduch M, Koide A, Lu V, Fischer J, Hurt E, Koide S, Kossiakoff AA, Hoelz A. (2015). Architecture of the fungal nuclear pore inner ring complex. *Science* 350, 56-64.

Fisher J, Teimer R, Amlacher S, Kunze R, Hurt E. (2015). Linker nups connect the nuclear pore complex inner ring with the outer ring and transport channel. *Nat. Struct. Mol. Biol.* 22, 774-81.

We apologize for the oversight. These citations have been added.

(2) Crystallographic analysis. The R_{free} values of the Nup120(1-754)•VHH-SAN10•VHHSAN11 and of the Nup85(1-564)•Seh1•VHH-SAN2 crystal structures are somewhat high (35.9% and 34.7%, respectively). These high R_{free} values may of course be a consequence of the low resolutions of these structures (4.3Å and 4.2Å, respectively). However, given that these nucleoporin structures have been previously solved at much higher resolution, one would expect much lower R_{free} values. It may be worthwhile for the senior author to take another look at the space group assignments. I suspect that subgroups of the assigned high symmetry hexagonal space groups would actually yield lower R_{free} values. I suspect the 6-fold axis may actually be imperfect and breaks down to a 2-fold or 3-fold axis. Regardless of this minor issue, both crystal structures clearly pinpoint the nanobody binding site on the target nucleoporin surface. Likewise, the evidence in Fig.4 and Supplementary Fig.5 supports the conclusion that VHH-SAN10 cannot be resolved because it binds to an unstructured loop of Nup120(1-754).

Totally valid point and we were somewhat perplexed by these R values as well. From the scaling statistics though, nothing suggests that a lower space group would be more appropriate. There are not more rejected reflections comparing scaling in a lower subgroup. In fact, very few rejections in general. Also, merging statistics are very similar. For the Nup120 structure the additional complication is that we only have a 140 degree sweep of data due to the radiation sensitivity of these high-solvent crystals, which would generate an incomplete dataset when using lower symmetry. Our suspicion is that with high resolution data the space group may indeed be lower, but that is indistinguishable at the modest resolution we are unfortunately limited to. We also think that the poor density around the nanobodies reflects a considerable amount of disorder, which we cannot model for, therefore the R value has to be on the higher end because of it. As the reviewer pointed out, the conclusions about the nanobody binding positions should be clear, and that is the main take home result. The previously published nanobody-free structures remain the benchmark structures regarding resolution and completeness/correctness of models.

(3) Table 2. The authors are encouraged to report Ramachandran and rotamer geometry statistics, as well as ClashScore and MolProbity scores.

The table matches the standard format of Nature Communications.

(4) Line 203. The conclusion that “nanobody binding causes a change in conformation” in Nic96 compared to previously solved crystal structures could be made more precise by specifying that the conformational change could also be induced by the novel, nanobody-mediated crystal lattice.

Correct. We rephrased the paragraph.

REVIEWERS' COMMENTS

Reviewer #1 (Remarks to the Author):

Dear all,

It appears I was the only one to question the publication of this toolbox in a separate paper. And I hope that the acceptance of this paper in a high impact journal will not catalyze a long list of publications, just describing the generation of panels of (characterized) Nanobodies against a particular target without a biological story (unlike the accompanying paper).

Anyway, I made my point. And since the other referees did not raise this concern, I will not oppose to the publication of this 'toolbox' paper.

Jan